# DNA Mismatch Repair Proteins and BRAF V600E Detection by Immunohistochemistry in Colorectal Cancer Demonstrates Concordance with Next Generation Sequencing

**Joel Yambert** [1], **Leigh A. Henricksen** [1], **June Clements** [1], **Andrew Hannon** [1], **Alyssa Jordan** [1], **Shalini Singh** [1], **Katerina Dvorak** [1,*], **Colin C. Pritchard** [2] **and Eric Q. Konnick** [2]

1   Roche Tissue Diagnostics, Tucson, AZ 85755, USA
2   Department of Laboratory Medicine and Pathology, University of Washington, Seattle, WA 98195, USA
*   Correspondence: katerina.dvorak@roche.com

**Abstract:** Background and Aims: Multiple laboratory methods are used to screen patients with colorectal cancer (CRC) for mismatch repair (MMR) protein deficiency to identify possible Lynch syndrome patients. The goal of this study was to compare the agreement between ready-to-use immunohistochemistry (IHC) assays for MLH-1, PMS-2, MSH-2, MSH-6, and mutated BRAF at V600E and molecular methods in CRC cases. The inclusion of the *BRAF* V600E mutation testing is important for the identification of patients with sporadic CRC, as the *BRAF* V600E mutation is very rarely observed in patients with Lynch syndrome tumors. Methods: CRC cases were analyzed by ColoSeq^TM tumor sequencing assay and VENTANA MMR IHC Panel that included anti-MLH1, anti-PMS2, anti-MSH2, anti-MSH6, and anti-BRAF V600E antibodies. Additionally, CRC cases with MLH1 IHC loss were evaluated for *MLH1* promoter hypermethylation. Results: One hundred and eighteen cases were analyzed. The overall percent agreement (OPA) for each evaluated marker status compared to next-generation sequencing (NGS) exceeded 96%. Twenty-three cases were positive for the BRAF V600E mutation by IHC and NGS, and twenty cases showed loss of MLH1 protein and were positive for *MLH1* hypermethylation. Samples with loss of MMR protein expression by IHC demonstrated genetic and/or epigenetic alterations that were consistent with the observed protein expression patterns. Conclusions: The results of this study indicate that ready-to-use IHC assays can correctly identify the loss of MMR proteins and the presence of mutated BRAF V600E protein, supporting the utility of the VENTANA MMR IHC Panel as an aid to stratify patients with sporadic CRC vs. potential Lynch syndrome.

**Keywords:** mismatch repair deficiency; MMR; MSI; dMMR; pMMR

## 1. Introduction

Colorectal cancer (CRC) is the third most common cancer and the fourth most prevalent cause of death in the world [1]. Approximately 15% of CRC develop through a pathway characterized by defective function of the DNA mismatch repair (MMR) system. CRCs with MMR deficiency (dMMR) are often poorly differentiated with proximal colon predominance, mucinous, medullary, or signet ring histologic features, and increased tumor-infiltrating lymphocytes [2,3].

DNA mismatch repair is a well-characterized pathway responsible for the identification and removal of mismatched bases in DNA [4]. The MMR proteins recognize the presence of a DNA mismatch and coordinate its removal and repair. In the absence of MMR activity, the accumulation of errors can lead to an increase in small insertions or deletions in DNA microsatellite regions, a phenotype described as microsatellite instability (MSI). The accumulation of mutations in cancer-related genes may lead to the development of malignancies. MMR deficiency may be caused by pathogenic germline or somatic variants

in one of the *MMR* genes (*MLH1*, *PMS2*, *MSH2*, or *MSH6*), where both alleles are mutated, or by epigenetic inactivation of the *MLH1* via methylation [5].

Lynch syndrome is defined as an autosomal dominant hereditary predisposition to colorectal cancer and other malignancies, including cancers of the endometrium, ovary, stomach, hepatobiliary tract, upper urinary tract, small bowel, pancreas, prostate, and brain as a result of a deleterious germline mutation in one of the MMR genes [6]. Approximately 5% of all CRCs are caused by Lynch syndrome [7] and can be early onset, with synchronous or metachronous neoplasms, although later presentations have also been described [8].

Current professional guidelines recommend universal dMMR screening of all colorectal cancer specimens, either by IHC or by MSI, for potential Lynch syndrome to identify patients and families that will benefit from further genetic testing and counseling [9–13]. IHC methods evaluate for loss of protein expression in formalin-fixed paraffin-embedded (FFPE) tissues, which can help target specific genes for further germline sequencing [14,15], while MSI testing identifies unstable microsatellite tracts resulting from dMMR. However, MSI testing is unable to identify directly the relevant mutated MMR protein [16–18]. Furthermore, although MSI and IHC are sensitive to Lynch syndrome, these methods are not specific, because about 12% or more of sporadic colorectal cancers exhibit loss of MLH1, due to hypermethylation of the *MLH1* promoter [6,19]. Importantly, it was found that the *BRAF* V600E mutation is observed in about two-thirds of tumors with loss of MLH1 expression or *MLH1* promoter methylation. However, the *BRAF* V600E mutation is very rarely observed in Lynch syndrome tumors [9,20,21].

To date, only a few large, controlled studies demonstrating concordance between IHC and sequencing have been performed with a standardized, automated MMR IHC panel on automated staining platforms [22–24]. However, the BRAF V600E status was not assessed in the cases included in these reports. This study evaluated the agreement between MMR protein expression and positive/negative BRAF V600E status in FFPE CRC samples using the standardized and automated VENTANA MMR IHC Panel and matched clinical-grade NGS and *MLH1* hypermethylation testing.

## 2. Methods

Study design: Tissues from 126 CRC cases were commercially obtained from Avaden Biosciences and evaluated by IHC for the expression of MMR and BRAF V600E proteins, by a clinically validated DNA sequencing assays designed to evaluate MMR genes and MSI status, and by *MLH1* hypermethylation analysis. DNA NGS interrogated genomic variants present in the MMR genes (*MLH1*, *PMS2*, *MSH2*, *MSH6*, *EPCAM*), *BRAF*, and 335 genes important in carcinogenesis (e.g., *POLE*, *POLD1*, *PIK3CA*, *KRAS*, *NRAS*, *ERBB2*, etc.) [25,26]. MMR sequencing included all exons, intronic and flanking sequences, and is capable of detecting single-nucleotide variants (SNVs), structural rearrangements (insertions, deletions, duplications, and translocations), and copy number variants (CNVs). MSI status was determined using the MSI-NGS method [27].

IHC testing was conducted at Roche Tissue Diagnostics (RTD, Tucson, AZ, USA). Sequencing and MSI testing was conducted at the University of Washington Genetics and Solid Tumors Laboratory (UW-GSTL, Seattle, WA, USA). *MLH1* hypermethylation testing was conducted by Mayo Clinic Validation Support Services (MVSS, Rochester, MN, USA).

Patient samples: This study intended to include at least 100 sequential CRC cases (not pre-screened) and additional enrichment CRC cases (pre-screened cases with a loss of at least one MMR protein). The cases used in the study were de-identified and unlinked to patient information. Sample selection criteria included properly fixed primary tumors with estimated ≥50% tumor content, ≥1 mm thickness, and acceptable morphology (Supplementary Table S1). Available information included age, gender, confirmed diagnosis, stage, and TNM stage. Additional cases with suspected loss of MMR proteins were selected to enrich for dMMR cases. All cases were blinded to all scientists and pathologists to any information that could link them to internal databases. Overall, 111 sequential cases and 15 enrichment cases were evaluated in this study.

Immunohistochemistry: The VENTANA MMR IHC Panel consists of VENTANA anti-MLH1 (M1) Mouse Monoclonal Primary Antibody, VENTANA anti-PMS2 (A16-4) Mouse Monoclonal Primary Antibody, VENTANA anti-MSH2 (G219-1129) Mouse Monoclonal Primary Antibody, VENTANA anti-MSH6 (SP93) Rabbit Monoclonal Primary Antibody, and VENTANA BRAF V600E (VE1) Mouse Monoclonal Primary Antibody. This panel is intended for the identification of individuals at risk for Lynch syndrome in patients diagnosed with CRC.

Ten slides from each case were divided into 5 sets. Each pair of slides was stained with one specific antibody from the VENTANA MMR IHC Panel and its corresponding respective monoclonal negative control antibody [Negative Control (Monoclonal) or Rabbit Monoclonal Negative Control] on a BenchMark ULTRA instrument using OptiView DAB IHC Detection Kit (RTD, Tucson, AZ, USA) according to manufacturer's instructions. After completion of staining, the slides were dehydrated and coverslipped according to the OptiView DAB IHC Detection Kit package insert.

All stained slides were scored by a trained RTD pathologist. MMR protein signal was classified as "Intact" or "Loss" based on nuclear localization in tumor cells only. Intact MMR protein expression was assigned to the case when unequivocal nuclear staining of any intensity above background was observed in viable tumor cells. Absence of any detectable signal or pale grey or tan nuclear discoloration in the viable tumor cells in the presence of internal positive controls (e.g., nuclear staining in lymphocytes, fibroblasts, or normal colonic epithelium) was considered as "Loss". The case was considered positive for BRAF V600E IHC when unequivocal cytoplasmic staining of any intensity above background was observed in tumor cells. Nuclear staining, weak to strong staining of isolated viable tumor cells/or small tumor clusters was considered negative [28]. In addition, morphology and background staining were evaluated in all slides.

Sequencing and MSI testing: The clinically validated ColoSeq™ Tumor assay [29] was performed by UW-GSTL personnel. Targeted NGS was performed on DNA extracted from ten unstained FFPE slides at 10 micron thickness from 126 samples. A matching H&E slide was assessed by a board-certified molecular- and anatomic-pathologist (EQK) for adequate tumor content and enrichment of neoplastic via macro dissecting. DNA was extracted as previously described [30], and DNA quality was assessed.

After DNA extraction, and NGS library preparation, ColoSeq™ Tumor testing was performed as previously described [29]. Briefly, DNA was sonicated to yield fragments of appropriate size for library preparation (Supplementary Table S2) using Hyperprep (Kapa Biosystems, Wilmington, MA, USA). Barcodes and sequence adapters were ligated to captured DNA fragments to allow for sequencing-by-synthesis using Illumina chemistries utilizing the NextSeq 500 (Illumina, San Diego, CA, USA) platform. Post-sequencing, data were de-multiplexed using standard bioinformatics tools. Alignment, variant calling, structural variant determination, copy-number analysis, MSI, and ethnicity determination were carried out using a clinically validated custom bioinformatics pipeline [25,26,31].

Average sequencing coverage was estimated using PICARD and CONTRA, with coverage less than $100\times$ considered suboptimal, and results were qualified with the caveat that false-negative results cannot be excluded. Data were considered to be a failed assessment based on a combination of overall coverage metrics, MSI loci coverage, loci-specific sequence depth, and assessment of overall sequencing quality, including manual review of selected sequences.

Neoplastic content was estimated from variant allele fraction (VAF) of neoplasm-associated variants and assessment of loss of heterozygosity (LOH) by haplotype analysis of cancer-associated mutations. Each sample was assessed for the presence of mutations in *BRAF* and other genes important in carcinogenesis (e.g., *PIK3CA*, *KRAS*, *NRAS*, *ERBB2*, etc.). Pathogenic and likely pathogenic mutations in MMR genes were enumerated, and a prediction of likely MMR IHC results was made based on the observed sequence variants. Samples with multiple mutations in a single MMR gene or a single mutation with LOH were considered likely to display loss of the appropriate protein product(s) on MMR IHC.

When variants in multiple MMR genes were identified, the genes with multiple alterations were generally deemed to have the most impact on protein expression, except in the cases where a BRAF p.V600E mutation was present. All samples were also assessed for variants in *POLE* and *POLD1* associated with hypermutation phenotypes, regardless of MSI status [32–34].

Pathogenic mutations were defined as variants that are likely to impact protein expression (e.g., stop gain, frameshift, or translocation mutations) or have been identified to be pathogenic/likely pathogenic in curated, publicly available databases, internal databases, or published literature [35]. Cases without identified significant MMR mutations were denoted as "No_pathogenic_variant_identified" and predicted to have intact MMR protein expression.

Additionally, MSI status [MSI-High (MSI-H), MSI-stable (MSS)] was determined for all samples using the clinically validated MSI-NGS method [27]. Samples with less than 60 evaluable loci were considered suboptimal, and samples with less than 28 evaluable loci were considered failures. A sample was considered MSI-H if at least 20% of evaluable loci were unstable. In cases of discrepancy between MMR IHC and MSI results, sequencing was considered the reference standard.

Methylation testing: All cases with reported loss of MLH1 protein by IHC were evaluated for *MLH1* hypermethylation [36] The methylation-specific PCR test included the control DNA from cell lines known to have an intermediate level (25%) or no hypermethylation. In addition, a system-level control (no DNA template) was included with all testing to verify the absence of any background signals.

Statistical analysis: The positive percent agreement (PPA) and negative percent agreement (NPA) analyses were the primary endpoint for all markers pooled. The individual markers had primary analyses centered on overall percent agreement (OPA) of each marker. Point estimates and 95% confidence intervals (CIs) were calculated for each marker and all markers pooled together. All analyses were conducted using SAS version 9.4 (SAS, Cary, NC, USA). For purposes of the study, the molecular status was used as the reference and classified as "Normal" or "Abnormal" based on sequencing, and accounting for *MLH1* hypermethylation results. After molecular status determination, IHC status was determined as "Correct" or "Incorrect" based on congruency between molecular status and IHC (Supplementary Table S3).

### 3. Results

Evaluable observations: A total of 126 cases, consisting of 111 sequentially selected cases and 15 enrichment cases were evaluated in this study. The total number of IHC observations was 630 (MLH1, PMS2, MSH2, MSH6, BRAF V600E) with paired sequencing analysis. Seven cases (35 IHC observations) failed sequencing. All slides stained with VENTANA MMR IHC Panel were acceptable except for four samples (3%) that failed IHC scoring criteria (unacceptable background, unacceptable morphology, tissue missing, etc.). Re-staining resulted in three samples passing IHC criteria, with one sample not acceptable due to poor morphology (five observations). Furthermore, two samples were excluded due to a lack of methylation results bringing the total to 588 evaluable observations used for the analysis (Figure 1).

Immunohistochemistry: Typical staining results were observed for CRC cases with proficient DNA mismatch repair status (pMMR, all four MMR proteins intact, Figure 2) or CRC cases with deficient DNA mismatch repair status (dMMR) such as sporadic CRC case shown in Figure 3 (loss of MLH1 and PMS2, intact MSH2 and MSH6 and presence of BRAF p. V600E). Potential Lynch syndrome cases with loss of MLH1 and PMS2 with negative BRAF V600E, or with loss of MSH2 and MSH6, are shown in Supplementary Figures S1 and S2.

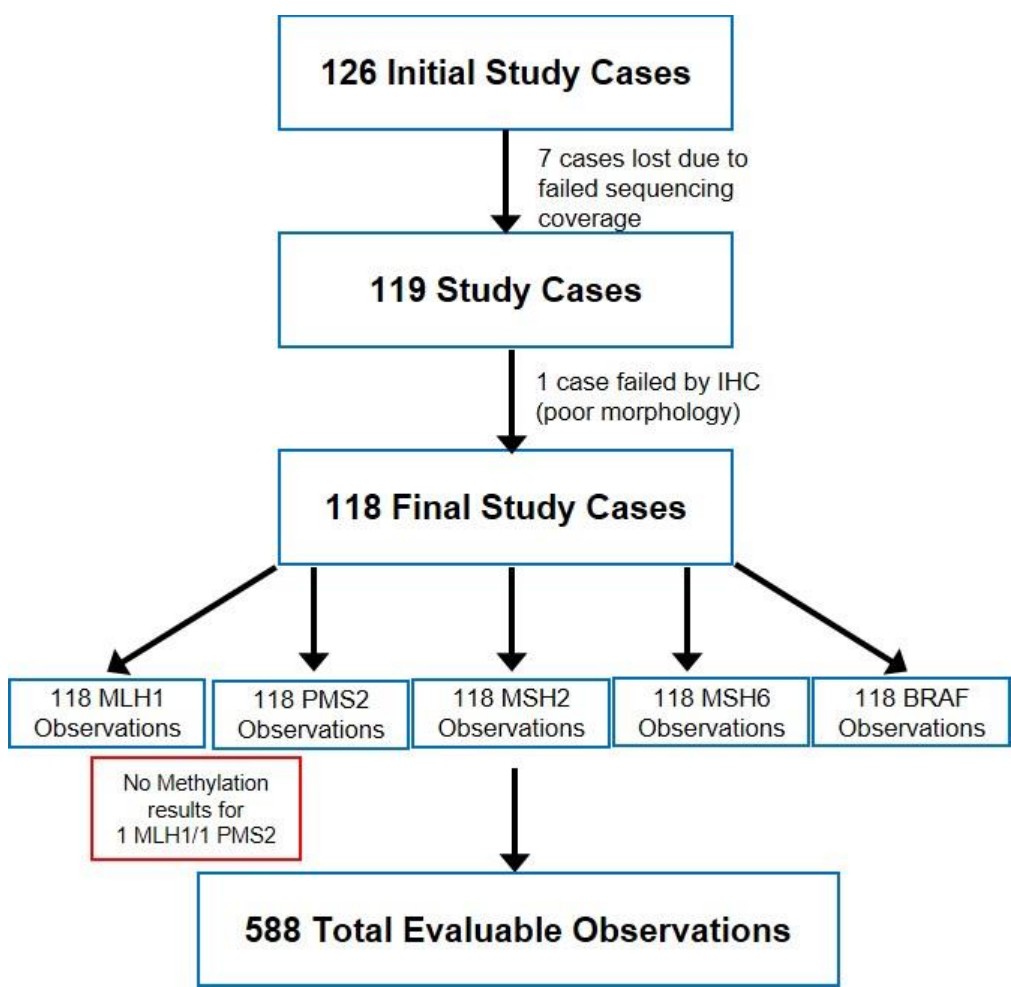

**Figure 1.** Study design and sample outcomes.

Pooled agreement between IHC and molecular testing for all MMR markers and BRAF V600E: IHC status was compared with results from molecular testing which included sequencing and hypermethylation testing (if appropriate) to determine pooled PPA and NPA for all four MMR markers and BRAF V600E. The point estimate for PPA and NPA was 99.4% (95% CI: 98.7, 100.0) and 93.5% (95% CI: 87.1, 98.6), respectively (Table 1). Additionally, the point estimate for OPA was 98.8% (95% CI: 98.0, 99.7).

**Table 1.** Primary pooled analysis for MMR markers plus BRAF V600E.

| IHC Test | Molecular Tests (i) | | | Agreement | | |
| --- | --- | --- | --- | --- | --- | --- |
| | Normal (ii) | Abnormal/Hypermethylated (iii) | Total | Type | n/N | % (95% CI) |
| Correct (iv) | 523 | 58 | 581 | PPA | 523/526 | 99.4 (98.7, 100.0) |
| Incorrect (iv) | 3 | 4 | 7 | NPA | 58/62 | 93.5 (87.1, 98.6) |
| Total | 526 | 62 | 588 | OPA | 581/588 | 98.8 (98.0, 99.7) |

(i) For MLH1, molecular testing included both sequencing data and promoter hypermethylation results. For PMS2, abnormal included sequencing data for *PMS2* and *MLH1* and hypermethylation as *PMS2* stability is linked to MLH1 status; for MSH2, molecular testing was solely based on the sequencing test; for MSH6, molecular testing included sequencing test results for *MSH6* and *MSH2* as MSH2 stability is linked to MSH6 protein status. For *BRAF* p.V600E, molecular testing was solely based on the sequencing test. (ii) For MMR markers, normal = molecular results support intact protein expression; for BRAF, normal = wild type *BRAF* (no V600E mutation). (iii) For MMR markers, abnormal = A sequencing result predicting loss of protein expression or positive for promoter hypermethylation; for BRAF, abnormal = presence of p.V600E mutation. (iv) A "Correct" or "Incorrect" IHC assessment was determined by viewing the congruency between molecular and IHC testing (see Supplementary Table S3).

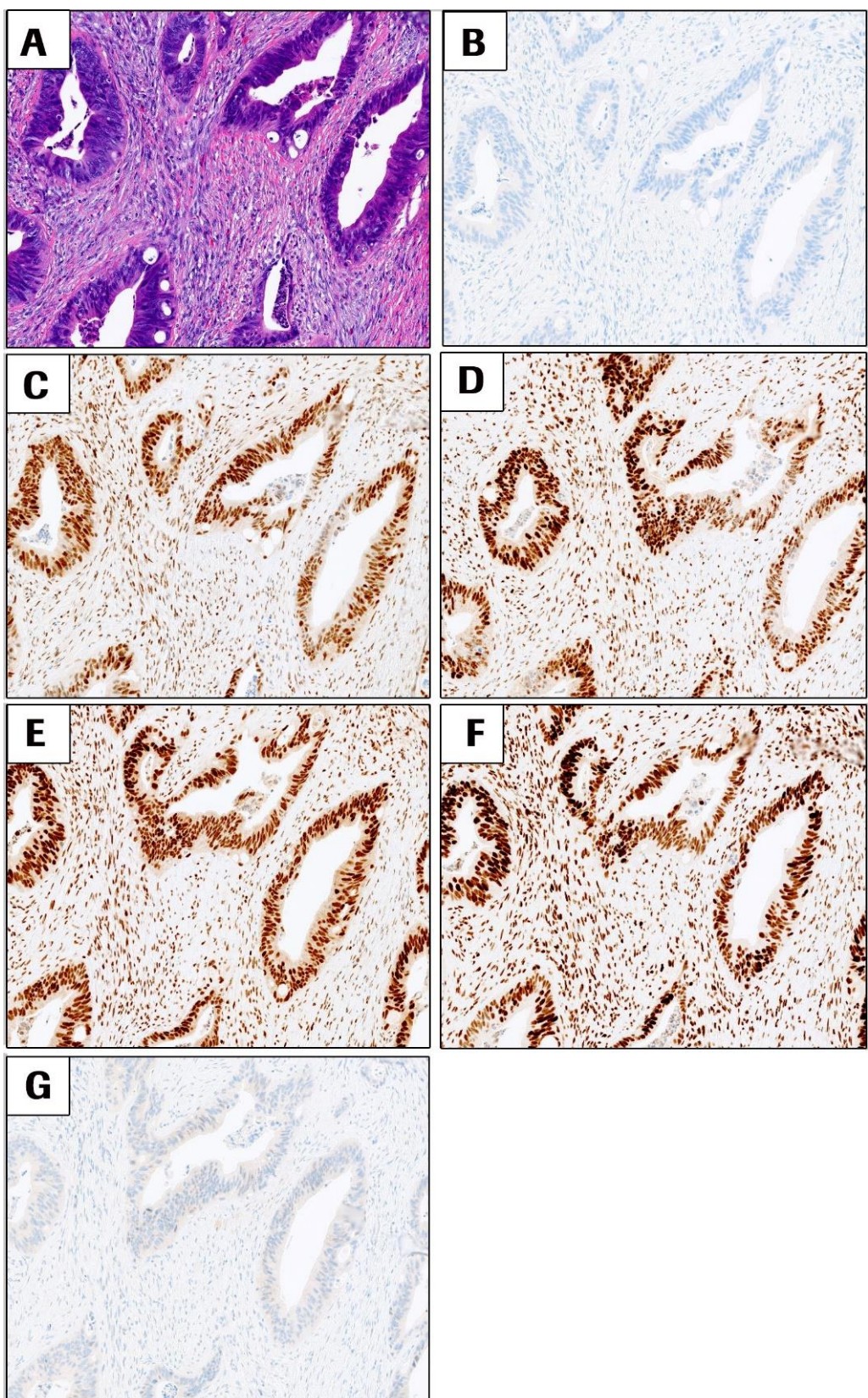

**Figure 2.** Representative images of a CRC case that exhibits proficient DNA mismatch repair status (pMMR) for all the four MMR markers in the VENTANA MMR IHC Panel and negative BRAF V600E. All images are at 10× magnification. (**A**) H&E, (**B**) negative control antibody, (**C**) MLH1, (**D**) PMS2, (**E**) MSH2, (**F**) MSH6, and (**G**) BRAF V600E staining. All images are at 10× magnification.

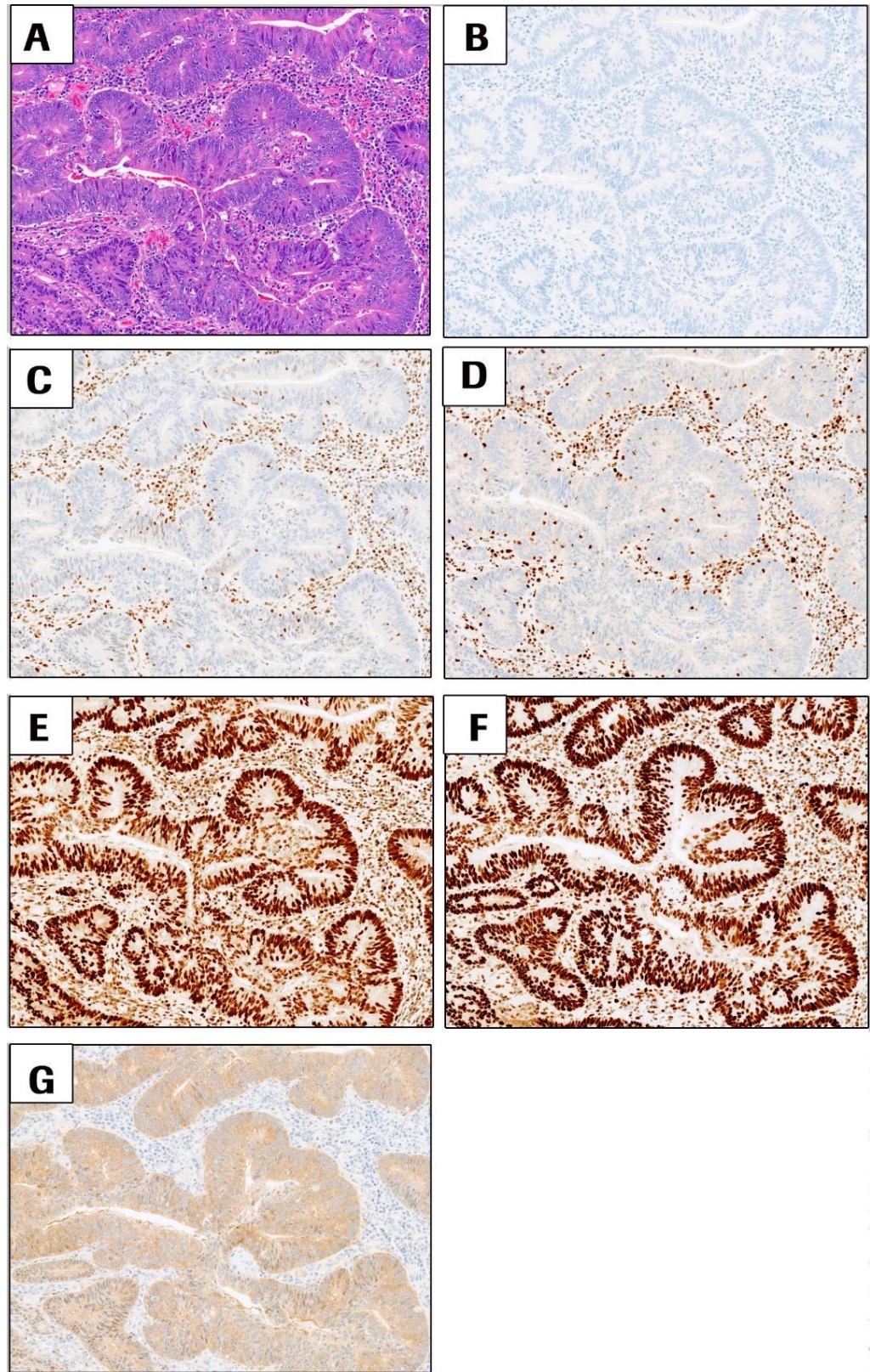

**Figure 3.** Representative images of sporadic CRC cases with deficient DNA mismatch repair status (dMMR) due to loss of MLH1 and PMS2 with intact MSH2 and MSH6. BRAF V600E is positive in this case. All images are at 10× magnification. (**A**) H&E, (**B**) negative control antibody, (**C**) MLH1, (**D**) PMS2, (**E**) MSH2, (**F**) MSH6, and (**G**) BRAF V600E staining (positive signal = brown cytoplasmic staining in tumor cells). All images are at 10× magnification.

MLH1 and PMS2 agreement between IHC and molecular testing: MLH1 or PMS2 IHC statuses were compared to results from molecular testing (NGS and hypermethylation) to determine the PPA and NPA for these two markers. OPA for MLH1 and PMS2 testing was 100.0% (95% CI: 96.8–100.0), and 99.2% (95% CI: 95.4–99.9), respectively (Table 2). PPA for MLH1 was 100.0% (95% CI: 96.0–100.0) and NPA 100.0% (95% CI: 87.1–100.0) while PPA for PMS2 was 99.1% (95% CI: 95.2–99.8) and NPA 100.0% (95% CI: 43.9–100.0). Twenty-six (26) cases were scored as MLH1 loss by IHC and identified as MSI-H, with twenty-four cases positive for *MLH1* promoter hypermethylation, supporting the observed loss of MLH1 protein expression. Two cases were negative for hypermethylation but were positive for somatic mutations in *MLH1* consistent with loss of protein expression. Due to the fact that PMS2 relies on MLH1 for protein stability, the PMS2 results take into account the MLH1 molecular and IHC status. Thirty (30) cases demonstrated PMS2 loss by IHC, of which twenty-six cases also showed MLH1 loss by IHC, while four cases demonstrated PMS2-only loss by IHC. Three (3) of these cases were MSI-H with pathogenic *PMS2* mutations predicted to result in loss of PMS2 protein, suggesting possible Lynch syndrome. One case had limited sequencing coverage and no mutations were identified in *PMS2,* but this case was MSI-H, which could be consistent with PMS2 loss by IHC. Discrepant cases are summarized in Supplementary Table S8.

**Table 2.** Agreement between IHC using anti-MLH1, anti-PMS2, anti-MSH2, and anti-MSH6 antibodies and NGS testing.

| Marker | Molecular Status/IHC Status | Agreement | | | |
| --- | --- | --- | --- | --- | --- |
| | | Type | n/N | % | 95% CI |
| MLH1 (i) | Normal/Intact | PPA | 92/92 | 100.0 | (96.0, 100.0) |
| | Abnormal/Loss | NPA | 25/25 | 100.0 | (86.7, 100.0) |
| | Total | OPA | 117/117 | 100.0 | (96.8, 100.0) |
| PMS2 (i) | Normal/Intact | PPA | 113/114 | 99.1 | (95.2, 99.8) |
| | Abnormal/Loss | NPA | 3/3 | 100.0 | (43.9, 100.0) |
| | Total | OPA | 116/117 | 99.2 | (95.3, 99.8) |
| MSH2 (ii) | Normal/Intact | PPA | 113/115 | 98.3 | (93.9, 99.5) |
| | Abnormal/Loss | NPA | 3/3 | 100.0 | (43.9, 100.0) |
| | Total | OPA | 116/118 | 98.3 | (94.0, 99.5) |
| MSH6 (ii) | Normal/Intact | PPA | 110/110 | 100.0 | (96.6, 100.0) |
| | Abnormal/Loss | NPA | 4/8 | 50.0 | (21.5, 78.5) |
| | Total | OPA | 114/118 | 96.6 | (91.6, 98.7) |

(i) For *MLH1*, molecular testing included both sequencing data and promoter hypermethylation results. For *PMS2*, abnormal included sequencing data for *PMS2* and *MLH1* and hypermethylation as PMS2 stability is linked to MLH1 status. (ii) For *MSH2*, molecular testing was solely based on the sequencing test. For *MSH6*, molecular testing included sequencing test results for *MSH6* and *MSH2* as MSH2 stability is linked to MSH6 status.

MSH2 and MSH6 agreement between IHC and molecular testing: The OPAs and 95% CIs for MSH2 and MSH6 were 98.3% (95% CI: 94.0–99.5), and 96.6% (95% CI: 91.6–98.7, Table 2), respectively. MSH2 PPA was 98.3% (95% CI: 93.9–99.5); MSH2 NPA was 100.0% (95% CI: 43.9–100.0). MSH6 PPA was 100.0% (95% CI: 96.6–100.0) and MSH6 NPA was 50.0% (95% CI: 21.5–78.5). Five cases demonstrated MSH2 loss and MSH6 loss by IHC. Sequencing results indicated that three cases contained mutations predicting protein loss, while the remaining two cases did not identify mutations predicted to lead to protein loss. Interestingly, these cases did have multiple genetic variants within the *MSH6* gene and could represent a rare loss of MSH2 due to MSH6 loss. Additional testing for Lynch syndrome would be appropriate for these cases in a clinical context.

Since MSH6 relies on MSH2 for protein stability, the MSH6 results take into account the *MSH2* molecular and IHC status. All five cases with MSH2 loss were scored as losses for MSH6 by IHC. Two of these cases contained multiple genetic variants within *MSH6*, which may have been the driver for both MSH6 and MSH2 loss. Two cases with only MSH6 loss had sequencing results with *MSH6* mutations predicted to result in MSH6 IHC loss, suggesting potential Lynch syndrome. In four cases, mutations were identified by sequencing that was predicted to result in MSH6 loss, but these cases were scored as intact by IHC. For case pUID00024 (Supplementary Table S8), the IHC staining was noted as heterogeneous with a focal region of strong staining. A review of NGS data noted that the *MSH6* mutation VAFs could be consistent with a subclonal population in the total tested area that could be consistent with the observed heterogeneous IHC staining. Case pUID00021 had intact protein expression not only for MSH6 but also for all markers and was MSS, but the sequencing revealed a hypermutated tumor with multiple *MSH6* variants, most likely the result of a *POLE* p.P286R mutation (Supplementary Table S8). The VAF of these variants suggested a compound heterozygous state in the tumor tissue, which could be associated with MSH6 loss. Similarly, two additional cases (pUID00023 and pUID00124) were predicted to show MSH6 loss based on sequencing data but showed MLH1/PMS2 loss by IHC (Supplementary Table S8). Both samples were positive for *MLH1* hypermethylation, which can cause mutations in other MMR genes identifiable by sequencing, such as in *MSH6*. In these cases, multiple sources of data are necessary to accurately interpret the underlying processes, how they relate to laboratory findings, and how to proceed with appropriate patient management.

BRAF V600E Agreement between IHC and NGS: This analysis compared the BRAF V600E IHC status to results from sequencing. Twenty-three (23) cases with positive BRAF V600E IHC were all positive for *BRAF* p.V600E via sequencing (Supplementary Table S4). The primary analysis for BRAF V600E IHC testing showed an OPA of 100% (95% CI: 96.8–100.0) when using the molecular testing result as the reference. The PPA was 100.0% (95% CI: 96.1–100.0) and NPA was 100.0% (95% CI: 85.7–100.0).

Additional testing was performed to verify the ability of the VENTANA anti-BRAF V600E (VE1) antibody to further stratify CRC cases showing a loss of MLH1 protein expression. Of the 23 positive BRAF V600E cases, 20 cases had a loss of MLH1 protein by IHC and were positive for MLH1 promoter hypermethylation. These data are consistent with the close association of BRAF V600E positive status with MLH1 promoter hypermethylation. The remaining three cases were pMMR (intact for all MMR proteins). Thus, all BRAF V600E-positive specimens were identified as sporadic CRC.

Enrichment and sequential case–cohort results were stratified by whether a case was part of the sequential or enrichment cohort. PPA, NPA, and OPA point estimates were calculated, along with 95% CIs for each cohort (Supplementary Table S5). The enrichment cohort had an OPA of 93.8% (95% CI: 87.7–98.5). PPA was 94.2% (95% CI: 87.5–100.0), and NPA was 92.3% (95% CI: 75.0–100.0). The sequential cohort had an OPA of 99.4% (95% CI: 98.7–100.0). The PPA was 100.0% (bootstrap CI not estimable) and NPA was 93.9% (95% CI: 86.7–100.0).

dMMR/pMMR IHC vs. molecular testing results: Overall MMR Panel results for IHC were compared to results of molecular testing. The presence of all MMR markers indicates that mismatch repair is proficient within the tumor (pMMR), while loss of any MMR protein is expected to lead to dMMR. In clinical practice, the complete MMR IHC Panel is considered when determining the MMR status of a given neoplasm. This analysis was completed by considering the status of the MMR proteins together as a panel to create a final dMMR/pMMR outcome for the two methods of comparison in the study (IHC vs. molecular testing). The dMMR/pMMR analysis showed an OPA of 97.5% (95% CI: 92.8–99.1), with a PPA of 98.8% (95% CI: 93.3–99.8) and an NPA of 94.7% (95% CI: 82.7–98.5). In total, there were three discrepant/discordant cases between IHC and molecular testing. Two cases (cases pUID00021 and pUID00024, Supplementary Table S8) were predicted dMMR by mutational analysis but were pMMR by IHC. Case pUID00021 was

MSS but hypermutated with pathogenic *POLE* and *MSH6* mutations. Case pUID00024 was hypermutated and MSI-high with multiple *MSH6* mutations, albeit with one mutation that suggested a subclonal population. Case pUID00042 was dMMR as indicated by IHC and MSI-H, but sequencing coverage was poor, and MMR gene mutation analysis was suboptimal.

Most dMMR cases with MLH1 protein loss are due to *MLH1* promoter hypermethylation. The inclusion of the BRAF V600E antibody to the MMR IHC Panel aids in the stratification of dMMR cases to differentiate sporadic CRC vs. potential Lynch syndrome. In this study, 24 of 26 CRC cases with MLH1 loss were positive for *MLH1* promoter hypermethylation. Twenty of the methylation-positive cases were identified positive for the BRAF V600E mutation by IHC, indicating that only six of the twenty-six cases with MLH1 loss would require additional *MLH1* hypermethylation testing to identify patients who would require additional Lynch syndrome screening. In these remaining MSI-H 6 cases, four cases were *MLH1* hypermethylation positive, supporting a sporadic etiology. Thus, only two cases (8%) were BRAF V600E and *MLH1* hypermethylation negative and would require additional workup for Lynch syndrome.

Discrepant cases: Results were considered discrepant if IHC clinical status was not in agreement with the predicted status as determined by sequencing analysis. Of the 588 IHC observations scored in the final study set, only seven discordant results for cases pUID00021, pUID00023, pUID00024, pUID00042, pUID00080, pUID00087, and pUID000124 were identified (Supplementary Table S8). In clinical practice, all of these cases would have been referred for additional testing. Case pUID00024 was the only case that was truly discordant. However, the focal intact MSH6 staining by IHC was consistent with the subclonal somatic mutation. Overall, most cases were correctly identified as pMMR or dMMR, and the discordance between individual IHC and sequencing results was technique related in origin.

## 4. Discussion

The objective of this study was to evaluate the concordance of MMR IHC and BRAF V600E IHC against molecular testing methods. Overall, percent agreement was excellent, with greater than 96% accuracy of all markers, strongly supporting the use of a standardized universal MMR IHC Panel to screen for dMMR in CRC in patients at risk for Lynch syndrome.

As loss of MLH1 protein can occur through multiple mechanisms, the inclusion of BRAF V600E (VE1) antibody helps to stratify CRC sporadic vs. potential Lynch syndrome cases with MLH1 loss. We observed that all CRC cases with *BRAF* p. V600E mutation were sporadic as confirmed by *MLH1* promoter hypermethylation or pMMR status testing. Four cases negative for *BRAF* p. V600E mutation with MLH1 loss were positive for *MLH1* hypermethylation, thus supporting a sporadic CRC origin. While MMR IHC and/or PCR-based MSI testing methods are common in diagnostic laboratories, testing for BRAF p. V600E mutation is not universally adopted in Lynch syndrome screening algorithms, but published studies suggest that BRAF V600E IHC is both accurate and cost-effective [37–40]. Thus, the inclusion of BRAF V600E IHC as an integrated part of a standardized Lynch syndrome screening approach provides valuable and rapid information to cases with MLH1 loss prior to *MLH1* methylation testing [20,41,42].

While seven possible discordant results out of the 588 IHC observations were identified (pUID00021, pUID00023, pUID00024, pUID00042, pUID00080, pUID00087, and pUID000124; Supplementary Table S8), analysis of the molecular results demonstrated biologically rational findings that highlight the need to integrate laboratory data from multiple modalities. In the case of pUID00021, the sample was MSS in the setting of intact MMR protein expression and negative BRAF V600E IHC. However, a *POLE* p.P286R mutation is associated with a "hypermutator phenotype" [29,32–34] resulting in multiple likely somatic *MSH6* mutations (1. p.E847*, 2. p.K1352T, 3, p.K1352T) at VAFs consistent with the heterozygous state in the neoplasm. Based on previous clinical experience, it

was predicted that the data were consistent with the loss of MSH6 protein expression. *POLE* hypermutation can result in multiple populations with heterogeneous and variable MMR IHC findings, even in the setting of multiple MMR gene mutations, regardless of MSI status. Interestingly, the *MSH6* p.K1352T mutation might be compatible with MSH6 expression, which would be consistent with the strong nuclear staining seen by IHC in this case. Overall, these data indicate that the patient is not at risk for Lynch syndrome, and the IHC and MSI results do not support additional clinical workup for consideration of that diagnosis.

The analysis of case pUID00023 showed MSI-H, loss of MLH1/PMS2, intact MSH2/MSH6, hypermethylated *MLH1* promoter, and no BRAF p.V600E mutation (Supplementary Table S8). Sequencing identified two *MSH6* variants (1. p.R1095H, 2. p.F1088Lfs*5). Although low VAF mutations were observed in *MLH1 (p.K196Nfs*6)* and *PMS2 (p.E109Gfs*30)* in the sequencing analysis, the assessment from sequencing data was that the *MSH6* variants could result in a loss of MSH6 protein expression. In consideration of the other molecular and IHC data, the observed MMR gene mutations were likely the secondary events of MSI-H resulting from MLH1/PMS2 loss due to *MLH1* hypermethylation, which is not directly detectable on the NGS assay used. These findings highlight the importance of integrating standard laboratory results with NGS data to enable appropriate interpretation.

In the case of pUID00024, MMR proteins were technically intact with negative BRAF V600E IHC; however, sequencing identified two mutations in *MSH6* (1. p.Q177*, 2. p.F1088Lfs*5) and predicted to result in MSH6 expression loss in the setting of MSI-H (Supplementary Table S8). Further investigation revealed heterogeneous MSH6 IHC expression, which included a strong region of positive nuclear staining, consistent with intraglandular/clonal heterogeneous expression. *MSH6* mutations VAFs (p.Q177*, 26% VAF, p.F1088Lfs*5, 14% VAF) were consistent with subclonal populations and congruent with the observed heterogeneous staining with focal areas of intact expression observed by IHC. These findings could be consistent with the multiple types of heterogeneous expression of MMR markers, including MSH6, between tumor areas and between tumor blocks that have been reported in the literature [43]. The observed findings are likely not related to Lynch syndrome; thus, additional studies and workups are necessary for that purpose. The implications of the observations for other indications, such as eligibility for immunotherapy are uncertain.

In the case of pUID00042, the IHC studies demonstrated only loss of PMS2 expression, while no *PMS2* alterations were identified by sequencing. Additionally, a heterozygous *MSH6* p.F1088Sfs*2 mutation was identified, which is insufficient to predict the loss of that protein. The tested sample yielded poor-quality DNA and the final sequencing data were limited due to low coverage. Nineteen (19) of 28 valid MSI loci were unstable, suggesting the tumor could be MSI-H. On review, PMS2 IHC loss was confirmed, although the staining of the internal controls was focal. Together, these data suggest this tissue may be suboptimal, possibly due to pre-analytic factors, preventing a definitive determination of the MMR status by either sequencing or IHC. Such cases should be carefully qualified in clinical testing and additional steps are recommended that may be necessary to prevent missed diagnostic opportunities.

Case pUID00080 was MSI-H, with mutations in *PMS2*, *MSH2*, and *MSH6*. MLH1/PMS2 was intact, while MSH2/MSH6 was lost by IHC. There was no *BRAF* p.V600E mutation detected by sequencing or IHC. The MSH2 (p.T320Yfs*13, 29% VAF) and *PMS2* (p.R628*, 31% VAF) mutations were both frameshifts, with the observed VAFs consistent with their presence at heterozygous frequencies in the neoplasm. Two variants were identified in *MSH6* at VAFs that were predicted to result in loss of MSH6 protein expression (1. p.S126P, 34% VAF, 2. p.F1088Lfs*5, 15% VAF). The identical situation was present in case pUID00087, which was MSI-H with mutations in *PMS2*, *MSH2*, *MSH6*, and IHC demonstrating loss of MSH2/MSH6 proteins. *MSH2* (p.T320Yfs*13, 27% VAF) and *PMS2* (p.R628*, 24% VAF) mutations were both frameshifts, with the observed VAFs consistent with their presence at heterozygous frequencies in the neoplasm. Two variants identified in *MSH6* were predicted

to result in loss of MSH6 protein expression (1. p.S126P, 31% VAF, 2. p.F1088Lfs*5, 17% VAF). Although these tissues were reportedly from separate anonymized cases, the overlap in all somatic mutations indicates that these two samples are from the same neoplastic clone from the same patient. Possible scenarios explaining the observed IHC and sequencing data include the presence of an undetected *MSH2* mutation(s), such as a structural mutation, or the observed IHC results may reflect a rare loss of MSH2 protein due to loss of MSH6 expression. Finally, *MSH2* methylation has been reported in CRC and has been suggested as a potential additional genetic alteration or "second hit" in Lynch syndrome [44], but testing for MSH2 methylation was beyond the scope of this study. Although the genetic etiology of dMMR was not identified in this case, all modalities correctly classified the sample as dMMR and would have correctly identified this patient for referral for additional clinical and laboratory evaluations.

Finally, case pUID00124 demonstrated loss of MLH1/PMS2, intact MSH2/MSH6, and also was negative for BRAF V600E by IHC. However, NGS identified two variants in *MSH6,* one of which was present at heterozygous germline frequency (p.K13T, 54% VAF) and the second being consistent with a somatic mutation (p.F1191Lfs*4, 26% VAF). The possible germline variant had been classified as a variant of uncertain significance in several databases, raising the possibility that this case could represent a patient with Lynch syndrome. This case was similar to case pUID00023 described above which was positive for *MLH1* hypermethylation, where the observed genetic variants were consistent with mutations secondary to MSI. Although the tissue for this case was exhausted during sectioning and there was insufficient tissue to resolve the *MLH1* promoter hypermethylation status by PCR-based method at MVSS, testing of residual DNA in UW-GSTL was positive for *MLH1* promoter hypermethylation. The available IHC and sequencing data are consistent with this tumor having a loss of MMR function, resulting in MSI, which would have correctly identified this patient for referral for additional clinical and laboratory evaluations, including *MLH1* hypermethylation testing.

Guidelines advocate universal screening of CRC tissues for Lynch syndrome [9,26,45]. However, barriers exist in the adoption of universal screening, including a lack of awareness, and variability in the implementation of guidelines, resulting in challenging workflow for this complex testing [46–48]. Key concerns of clinical laboratories include: (1) the time required to navigate through the decision points of a complex process; (2) the lack of a clear champion to obtain commitment and consensus for testing across departments; and (3) uncertainty as to which department(s) should "own" the screening program [46]. While germline sequencing is required for a definitive diagnosis of Lynch syndrome, this is not a practical screening approach due to the extensive time and cost involved, especially when referral to genetics counseling and medical genetics is considered. MMR testing by IHC enables an accessible and practical universal screening approach, at the time of histopathological diagnosis, that can guide subsequent decisions regarding definitive sequencing tests and also identifies the likely implicated gene in high-risk patients [49] (Figure 4). Alternative approaches such as MSI or MSI by NGS testing may require additional laboratory facilities and expertise and may be limited by biological factors such as limited tumor content or low levels of MSI observed in some cancers with *MSH6* germline mutations [50,51].

The VENTANA MMR IHC Panel is the first ready-to-use MMR IHC Panel on an automated staining platform that has been shown to be concordant with sequencing. Automation and availability of a ready-to-use panel from a single vendor will simplify workflow, and the inclusion of BRAF IHC as a surrogate for MLH1 hypermethylation accurately identifies cases with such somatic epimutations. The importance of an efficient and integrated solution will have an impact beyond the patient, as universal screening may allow for more comprehensive utilization of targeted genetic testing of at-risk family members to identify Lynch syndrome carriers for prevention of additional cancers, potentially reducing the burden of cancer in this population.

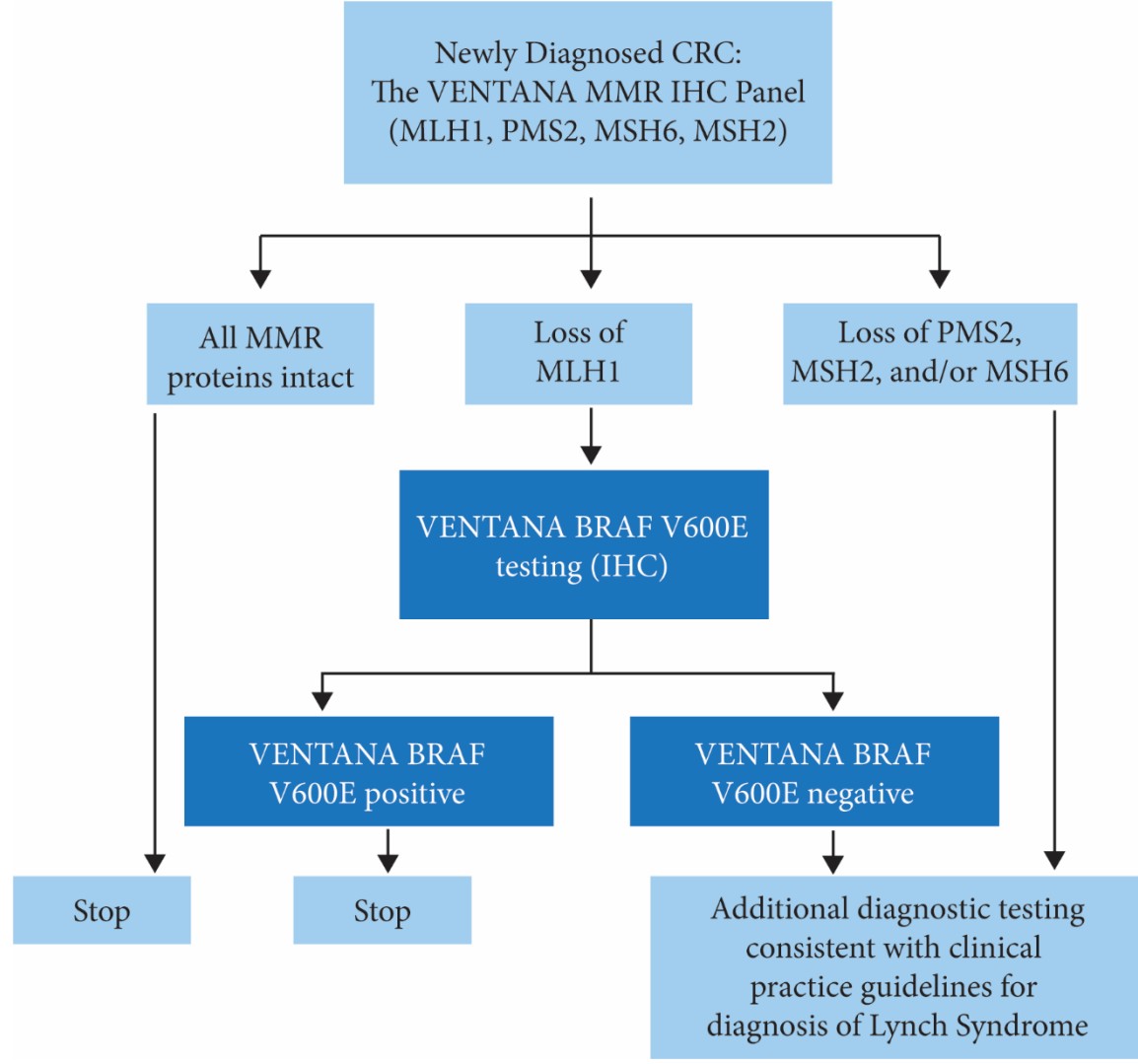

**Figure 4.** Proposed MMR testing algorithm for the identification of individuals at risk for Lynch syndrome in patients diagnosed with CRC. Additional testing of samples with MLH1/PMS2 loss and negative BRAF V600E could include MLH1 promoter hypermethylation testing and/or referral to medical genetics and/or genetic counseling for germline MMR gene testing.

**Supplementary Materials:** The following supporting information can be downloaded at: https://www. mdpi.com/article/10.3390/jmp3040029/s1, Figure S1: Representative images of possible Lynch syndrome CRC case with deficient DNA mismatch repair status (dMMR) due to MLH1 and PMS2 loss, with intact MSH2 and MSH6, BRAF V600E is negative for this case. Figure S2: Possible Lynch Syndrome case exhibiting deficient DNA mismatch repair status (dMMR) due to MSH2 and MSH6 loss, with intact MLH1 and PMS2, BRAF V600E is negative for this case. Table S1: Evaluation Criteria for Tissue for Study Enrollment. Table S2: Key Genomic Loci Sequenced in ColoSeq[TM]. Table S3: Molecular Status & IHC Assay Congruency. Table S4: Agreement between IHC using VENTANA anti-BRAF V600E (VE1) antibody and NGS Testing. Table S5: IHC vs. Molecular Testing Status for Enrichment & Sequential Cohort. Table S6: Pooled Analysis for VENTANA MMR IHC Panel Agreement between IHC and Molecular Testing. Table S7: Pooled Analysis for four MMR IHC markers (without VENTANA anti-BRAF V600E (VE1) antibody) Agreement between IHC and Molecular Testing. Table S8: Description of Discrepant Cases.

**Author Contributions:** Conceptualization, E.Q.K. and L.A.H.; methodology, J.Y., L.A.H., C.C.P. and E.Q.K.; validation, K.D. and L.A.H.; formal analysis, J.Y., C.C.P., S.S., J.C., C.C.P. and E.Q.K.; data curation, J.C., A.J., C.C.P., A.H. and E.Q.K.; writing—original draft preparation, K.D. and E.Q.K.; writing—review and editing, J.Y., L.A.H. and C.C.P. Supervision, L.A.H., S.S. and E.Q.K. All authors have read and agreed to the published version of the manuscript.

**Funding:** Funded by RTD.

**Institutional Review Board Statement:** Ethical review and approval were waived for this study because all CRC tissues used in the study were commercially obtained from Avaden Biosciences.

**Informed Consent Statement:** Patient consent was waived since all tissues used in this study were obtained from Avaden Biosciences, they were de-identified and unlinked to patient information.

**Data Availability Statement:** The data presented in this study are available on request from the corresponding author.

**Conflicts of Interest:** Leigh A. Henricksen, June Clements, Andrew Hannon, Alyssa Jordan, Shalini Singh, and Katerina Dvorak are employees of Roche Tissue Diagnostics.

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
