# Peer review of "DNA Mismatch Repair Proteins and BRAF V600E Detection by Immunohistochemistry in Colorectal Cancer Demonstrates Concordance with Next Generation Sequencing"

_jmp, doi:10.3390/jmp3040029_

Round 1

Reviewer 1 Report (Previous Reviewer 2)

This is an important and interesting study. 

Author Response

There are no comments that require the response. The manuscript was checked for the spelling errors. 

Reviewer 2 Report (New Reviewer)

The primary objective of this study is to evaluate concordance of MMR and BRAF V600E IHC analysis against standard molecular testing methods. The study finds overall excellent agreement with >96% accuracy of all the markers. These results support the use of a standardized universal ready-to-use MMR IHC panel (commercially available from Ventana) to screen for dMMR in CRC patients at risk for LS. An important aspect to this work is that BRAF IHC was able to accurately stratify CRC vs. potential LS cases that had MLH1 loss, adding this approach to existing screening strategies for LS. Overall, there sufficient data provided to support the conclusions. Seven possible discordant results out of the total of 588 HIC observations were identified and they are discussed in detail. 

­­

Introduction

Para 4 - The last sentence isn't clear - 'MSI testing is unable to directly identify the relevant MMR protein and decreased reliability in biopsy specimens vs. IHC'.

Para 5 - Not clear how this paragraph fits into the flow of the introduction?

para 6- If three large studies have already been published, it isn't clear why the addition of BRAF IHC analysis provides significant novelty to this work. Perhaps the novelty of this study could be better articulated.

Methods

Why was MLH1 hypermethylation status assayed in this study?

Is DNA NGS and massively-parallel DNA sequencing used interchangeably?

The methods section is difficult to follow in terms of technologies applied for each endpoint - e.g., how was MLH1 hypermethylation testing accomplished?

The authors should provide a citation for 'pathogenic mutations' since this is a rapidly evolving field of research.

Reference for methylation testing is needed

Results

Figures 2,3 - The negative control antibody is provided for which antigen? This is not really critical data and could be provided in supplemental, if at all. Also, it is difficult to assess the BRAF positive staining in Figure 3. Perhaps the positive staining can be indicated by an arrow. This would also be a place to include a negative control.

Discussion

Although the role of BRAF mutation testing to distinguish sporadic from LS CRCs is mentioned, the importance of assessing this marker for sporadic CRC should be included in the Abstract and Introduction with an appropriate citation provided (e.g., Blaker et al., 2020, Int. J Cancer)

Author Response

Response to reviewer

Introduction

Para 4 - The last sentence isn't clear - 'MSI testing is unable to directly identify the relevant MMR protein and decreased reliability in biopsy specimens vs. IHC'.

We agree with this comment and thus this sentence was deleted from the manuscript.

Para 5 - Not clear how this paragraph fits into the flow of the introduction?

We agree with this comment. This paragraph was deleted from the manuscript and replaced with the sentances explaining the role of MLH1 promoter methylation and BRAF V600E mutation on CRC.

Para 6- If three large studies have already been published, it isn't clear why the addition of BRAF IHC analysis provides significant novelty to this work. Perhaps the novelty of this study could be better articulated.

Thank you for your comment. Although three studies were published on the concordance between NGS and IHC methods, none of these studies included the evaluation of the BRAF V600E mutation. The detection of the V600E BRAF mutation in an MSI-H CRC is evidence against the presence of a germline mutation in either MLH1 or MSH2 (Shia, 2015). This test is important for identification of patients with hereditary vs. sporadic CRC. As the more explanation on the role of BRAF V600E mutation was included in paragraph 4, we think that the introduction is now more clear.

Methods

Why was MLH1 hypermethylation status assayed in this study?

The main reason why the hypermethylation of the MLH1 promoter was evaluated in this study is that 12% or more of sporadic colorectal cancers exhibit loss of MLH1, due to hypermethylation of the MLH1 promoter (Boland, 2010, Herman, 1998; Shia, 2015) and subsequent loss of MLH1 expression. These patients exhibiting MLH1 promoter hypermethylation have sporadic cancer. We included the explanation in the revised manuscript including appropriate references (page 2).  

Is DNA NGS and massively-parallel DNA sequencing used interchangeably?

We agree, it is confusing, the sentence containing “massively-parallel DNA sequencing” was modified and “massively-parallel” was deleted (page 2).

The methods section is difficult to follow in terms of technologies applied for each endpoint - e.g., how was MLH1 hypermethylation testing accomplished?

The evaluation of the MLH1 promoter hypermethylation was done using methylation specific PCR test. This test was performed and at the Mayo Clinic, Department of Laboratory Medicine and Pathology in accordance with CAP/CLIA requirements and/or standard laboratory practices. A PCR-based assay is used to test tumor DNA for the presence of hypermethylation of the MLH1 promoter. This is a modification of the method described by Grady et al.(Grady WM, Rajput A, Lutterbaugh JD, Markowitz S: Detection of aberrantly methylated hMLH1 promoter DNA in the serum of patients with microsatellite unstable colon cancer. Cancer Res 2001;61:900). The test included the addition of control DNA from cell lines known to have an intermediate level (25%) or absence of hypermethylation. In addition, a system-level (no DNA template) was included with all testing to verify the absence of any background signals. The reference was included (page 4).

The authors should provide a citation for 'pathogenic mutations' since this is a rapidly evolving field of research.

The reference for this section was added (He, 2016, page 4).

Reference for methylation testing is needed.

The reference for methylation testing was added (Grady, 2011, page 4).

Results

Figures 2,3 - The negative control antibody is provided for which antigen? This is not really critical data and could be provided in supplemental, if at all. Also, it is difficult to assess the BRAF positive staining in Figure 3. Perhaps the positive staining can be indicated by an arrow. This would also be a place to include a negative control.

In Figure 3, the brown cytoplasmic staining that is indicative of presence of BRAF V600E mutation in the majority of tumor cells in the panel G, therefore it could be confusing to include few arrows in the image. Instead of arrows the following comment was added to the Figure legend: (positive signal =brown cytoplasmic staining in tumor cells).

The images of tissues stained with negative reagent control (NRC) are showing no specific signal, they are nearly identical, regardless if they were stained using Rabbit Monoclonal Negative Control Ig (P/N 790-4795) or Negative Control (Monoclonal) (PN 760-2014, mouse). Therefore, only one image was included in each Figure. 

Discussion

Although the role of BRAF mutation testing to distinguish sporadic from LS CRCs is mentioned, the importance of assessing this marker for sporadic CRC should be included in the Abstract and Introduction with an appropriate citation provided (e.g., Blaker et al., 2020, Int. J Cancer)

The Introduction and Abstract were modified. The importance of BRAF V600E mutation testing is now  included in the Introduction (Paragraph 4) and the Abstract.

The manuscript was rechecked for spelling errors.

This manuscript is a resubmission of an earlier submission. The following is a list of the peer review reports and author responses from that submission.

Round 1

Reviewer 1 Report

The objective of this study was to compare the agreement between ready-to-use immunohistochemistry (IHC) assays and molecular methods for the screening of colorectal cancer (CRC) patients for mismatch repair (MMR) protein deficiency, to identify possible Lynch syndrome patients. The results of this study showed that the overall percent agreement between the two methods for each evaluated marker is not inferior to 96%. The results of this study support the view that ready-to-use IHC assays can correctly identify loss of MMR proteins.

Although these results are of interest, their originality is questionable. The authors in the Introduction stated that “to date, no large, controlled studies demonstrating concordance between IHC and sequencing have been performed with a standardized, automated MMR IHC panel on an automated staining platform from a single vendor”. However, probably the authors of the present study have probably missed some of the recent literature on this topic: see Malapelle U et al. Evaluation of microsatellite instability and mismatch repair status………….   Cells 2021; 10: 1878; Amemiya K et al. Simple IHC reveals complex MMR…………………….   Cancer Medicine 2022 ; in press.

The authors do not discuss the potential pitfalls related to MMR evaluation by IHC. These limitations of MMR deficiency by IHC are analyzed in some recent studies, such as McCarthy A et al. J Pathol Clin Res 2019; 5: 115-129.

In conclusion, despite the interesting results observed in this study, its originality is limited.

Author Response

Thank you for your comments. Three missing additional references were added to the manuscript (References 19-21). Additionally, the paragraph was modified: "To date, only a few large, controlled studies demonstrating concordance between IHC and sequencing have been performed with a standardized, automated MMR IHC panel on an automated staining platform from a single vendor 19-21. However, none of these studies evaluated the cases' BRAF V600E status.  

Reviewer 2 Report

The manuscript “DNA Mismatch Repair Proteins and BRAF V600E Detection by Immunohistochemistry in Colorectal Cancer Demonstrates Concordance with Next Generation Sequencing” by Yambert et al. compares different screening methods of colorectal cancer to identify Lynch Syndrome risk based on mismatch repair protein deficiencies. The authors identified that using ColoSeqTM tumor sequencing assay and VENTANA MMR IHC Panel on 118 patient samples resulted in more than 96% overall precent agreement for each evaluated marker indicating that ready-to-use IHC assays can correctly identify loss of MMR proteins and the presence of mutated BRAF V600E protein.

Strength:

This comprehensive study supports the utility of the VENTANA MMR IHC Panel to identify loss of MMR proteins and the presence of mutated BRAF V600E protein as an aid to stratify patients with sporadic 24 CRC vs. potential Lynch syndrome. These data are impactful as screening such as the VENTANA MMR IHC Panel can provide an efficient and accurate technique to target genetic testing of at-risk Lynch Syndrome carrier family members.

Minor comments:

The authors should clarify for the general research audience what “sequentially selected cases” and “enrichment cases” means as this as an important criteria for the case cohort selection.

On occasion the formatting seems to be off for font and size, e.g., page 1 line 43 MHL1 and page 2 line 48, Lynch Syndrome

Author Response

Thank you for you comments. The following sentence was added to the manuscript to fully explain the difference between sequential and enriched cases. "This study intended to include at least 100 sequential CRC cases (not pre-screened) and additional enrichment CRC cases (pre-screened cases with a loss of at least one MMR protein)." The formatting was fixed. 

Reviewer 3 Report

The paper is presented in too simple way.  The author should add all data (the slides) for all the patients (maybe in supplementary) not only limiting to a representative slides. Staining quantification could be reported as graphic.The author should think how represent they data in better way using graphic and not only tables.

Author Response

This review has absolutely no value. It is not specific, apparently done in rush. I have never seen any paper showing images for 118 cases (it means close to 600 images would have to be shown).  Please, do not use this reviewer for the reviewing manuscripts.